# Magnetic-field-dependent quantum emission in hexagonal boron nitride at room temperature

Annemarie L. Exarhos [1,4], David A. Hopper [1,2], Raj N. Patel [1], Marcus W. Doherty [3] & Lee C. Bassett [1]

Optically addressable spins associated with defects in wide-bandgap semiconductors are versatile platforms for quantum information processing and nanoscale sensing, where spin-dependent inter-system crossing transitions facilitate optical spin initialization and readout. Recently, the van der Waals material hexagonal boron nitride (h-BN) has emerged as a robust host for quantum emitters, promising efficient photon extraction and atom-scale engineering, but observations of spin-related effects have remained thus far elusive. Here, we report room-temperature observations of strongly anisotropic photoluminescence patterns as a function of applied magnetic field for select quantum emitters in h-BN. Field-dependent variations in the steady-state photoluminescence and photon emission statistics are consistent with an electronic model featuring a spin-dependent inter-system crossing between triplet and singlet manifolds, indicating that optically-addressable spin defects are present in h-BN.

[1] Quantum Engineering Laboratory, Department of Electrical and Systems Engineering, University of Pennsylvania, Philadelphia, PA 19104, USA. [2] Department of Physics and Astronomy, University of Pennsylvania, Philadelphia, PA 19104, USA. [3] Laser Physics Centre, Research School of Physics and Engineering, Australian National University, Canberra, ACT 2601, Australia. [4] Present address: Department of Physics, Lafayette College, Easton, PA 18042, USA. Correspondence and requests for materials should be addressed to L.C.B. (email: lbassett@seas.upenn.edu)

Spins in semiconductors are the elementary units for quantum spintronics[1], enabling an array of technologies, including quantum communication[2,3], spin-based nanophotonics[4], nanoscale nuclear magnetic resonance[5,6], and in vivo transduction of intracellular magnetic, thermal, and chemical fields[7]. Leading candidates in diamond[8–10] and silicon carbide[11,12] exhibit room-temperature, spin-dependent photoluminescence (PL) that facilitates initialization and readout of individual electron spins, along with their proximal nuclear spins[13]. Substantial progress notwithstanding, synthesis and device fabrication with these three-dimensional semiconductors remains challenging, especially for sensing applications, which demand the use of near-surface spins whose quantum properties are degraded as compared to the bulk. Intrinsically low-dimensional materials, such as the van der Waals material hexagonal boron nitride (h-BN), offer an appealing alternative—spins confined to the same two-dimensional (2D) atomic plane, and all at the surface, offer enormous potential to engineer quantum functionality.

Magneto-optical effects are the principal means by which individual spins are addressed[14] and coupled to light[15]. Quantum emission in h-BN[16–22] is believed to originate from defects with localized electronic states deep within its bandgap, similarly to other wide-bandgap materials exhibiting defect-related single-photon emission[23]. However, even in this expanding catalog of materials and their numerous fluorescent defects[24], room-temperature, spin-dependent PL remains a rare phenomenon due to the necessary alignment of energy levels and symmetry-protected selection rules. Despite well-established electron paramagnetic resonance signatures for bulk h-BN[25,26] and recent theoretical predictions[27], experimental evidence for magneto-optical effects has been elusive to date[28,29].

Present understanding of the chemical and electronic structure of h-BN's quantum emitters (QEs) is impeded by the heterogeneity of their optical properties[21]. Contending models aim to account for disparate observations; multiple QE species likely play a role[16,30–32]. Nonetheless, h-BN's QEs universally exhibit linearly polarized optical absorption and emission consistent with optical dipole transitions from a defect with broken in-plane symmetry[16,21,22]. Based on symmetry considerations, any spin-dependent inter-system crossing (ISC) transitions likely produce an anisotropic PL response to in-plane magnetic fields. Here, we exploit that fact to identify and characterize individual QEs in h-BN with spin-dependent optical properties. We demonstrate that select QEs in h-BN do exhibit room-temperature, magnetic-field-dependent PL consistent with a spin-dependent ISC, paving the way to the development of 2D quantum spintronics.

## Results

**Identification of magnetic-field-dependent QEs**. We study a 400-nm thick exfoliated h-BN flake suspended across a set of holes etched in a silicon substrate at room temperature in ambient conditions [Fig. 1a]. Excitation–polarization-resolved PL images [Fig. 1b] reveal a number of strongly linearly polarized emitters in the suspended region. Unfortunately, the absence of well-characterized defect-specific emission signatures prevents the selective addressing of defect subensembles, as has been essential for statistical studies and the identification of spin qubits in other materials[14,33]. Consequently, we study QEs in h-BN at the single-defect level. To identify individual magnetic-field-dependent emitters, we construct differential images [Fig. 1c] of the PL variation, $(I_B - I_0)/I_0$, where $I_B$ ($I_0$) is the brightness extracted from composite PL maps with (without) a magnetic field, **B**, applied along the horizontal in-plane direction (see Methods).

Most of the emitters in the suspended region [below the dashed curve in Fig. 1c] show no change with the magnet, whereas a few

red and blue features highlight potentially interesting spots. Upon further study, some features are not reproducible, but two emitters in particular, denoted by circles and squares in Fig. 1b, c, exhibit systematic changes in brightness due to the applied field. Strikingly, the circled emitter brightens whereas the boxed emitter dims in response to **B** at this orientation. The following discussion focuses on the circled defect, which remained stable over several months. Data for the boxed emitter and other field-dependent spots are also available in Section IF of the Supplementary Information.

Figure 1d–f summarizes the spatial, temporal, and spectral emission characteristics of the QE circled in Fig. 1b, c. Like many QEs in h-BN[19,21,22], the PL exhibits incomplete visibility as a function of linear excitation–polarization angle, with an optimum excitation axis (hereafter called the absorptive dipole) offset from the fully polarized emission-dipole axis by an angle $\Delta = 53° \pm 4°$. The absorptive dipole orientation is independent of **B**; see Supplementary Figure 4. The background-corrected second-order autocorrelation function, $\tilde{g}^{(2)}(t)$, [Fig. 1e] exhibits an antibunching dip near zero delay that drops below the threshold, $\tilde{g}^{(2)}(0) < 0.5$, indicating the PL is dominated by a single emitter (see Methods). Figure 1f shows the QE's room-temperature PL spectrum with and without an applied magnetic field.

In Fig. 1, the absorptive dipole of the circled QE is horizontal, || **B**. As illustrated in Fig. 1g, we explore arbitrary field orientations by rotating the sample about the optical axis, where $\alpha$ ($\varepsilon$) denotes the orientation of the absorptive (emissive) dipole in the plane of the sample ($\hat{x}$-$\hat{y}$ plane), relative to $\hat{x}$, and by adjusting a magnet goniometer in the $\hat{x}$-$\hat{z}$ plane (out of the sample plane), where $\beta$ is the angle of the field relative to $\hat{x}$.

**Variations in steady-state PL**. Figure 2a shows the PL variation as a function of sample orientation when an 890 G magnetic field is applied along $\hat{x}$ ($\beta = 0°$). The dashed line denotes the zero-field emission rate. The QE exhibits both increased and decreased emission as a function of the in-plane field direction, with >50% variation in both directions. Furthermore, the 90° modulation period is approximately aligned to the optical dipole orientations, such that the PL is brighter (dimmer) when the field is either aligned or perpendicular to the absorptive (emissive) dipole. The observed 90° periodicity persists when varying magnetic field strength [Figs. 2a and 3c, for example]. While this anisotropic PL modulation is reminiscent of other quantum emitters with spin-dependent ISC transitions[34], the 90° symmetry and bipolar response (i.e., both brightening and dimming) are unique.

The disparate behavior as a function of $B$ for different sample orientations is illustrated in Fig. 2b. The PL increases monotonically with $B$ at $\alpha = 90°$ whereas the response at $\alpha = 45°$ is nonmonotonic; an initial increase out to ≈70 G is followed by decreasing PL which eventually falls below the zero-field emission rate. In both orientations, the variation appears to saturate by ≈600 G and is independent of optical excitation power; see Supplementary Figure 3.

For out-of-plane fields ($\beta = 90°$), the PL increases monotonically [Fig. 2c], although the variation saturates by ≈200 G and is noticeably smaller than for $\beta = 0°$. The offset between data sets at different sample orientations likely reflects uncertainty in estimating the zero-field emission rate. This behavior, along with observations of a similar monotonic increase observed for $\beta = 45°$ [Fig. 2d], suggests an underlying 180° symmetry for rotations about $\hat{x}$ or $\hat{y}$, contrasting with the 90° periodicity observed for rotations about $\hat{z}$.

**Photo-dynamic response**. The QE's photon emission statistics provide insight into the field-dependent optical dynamics that

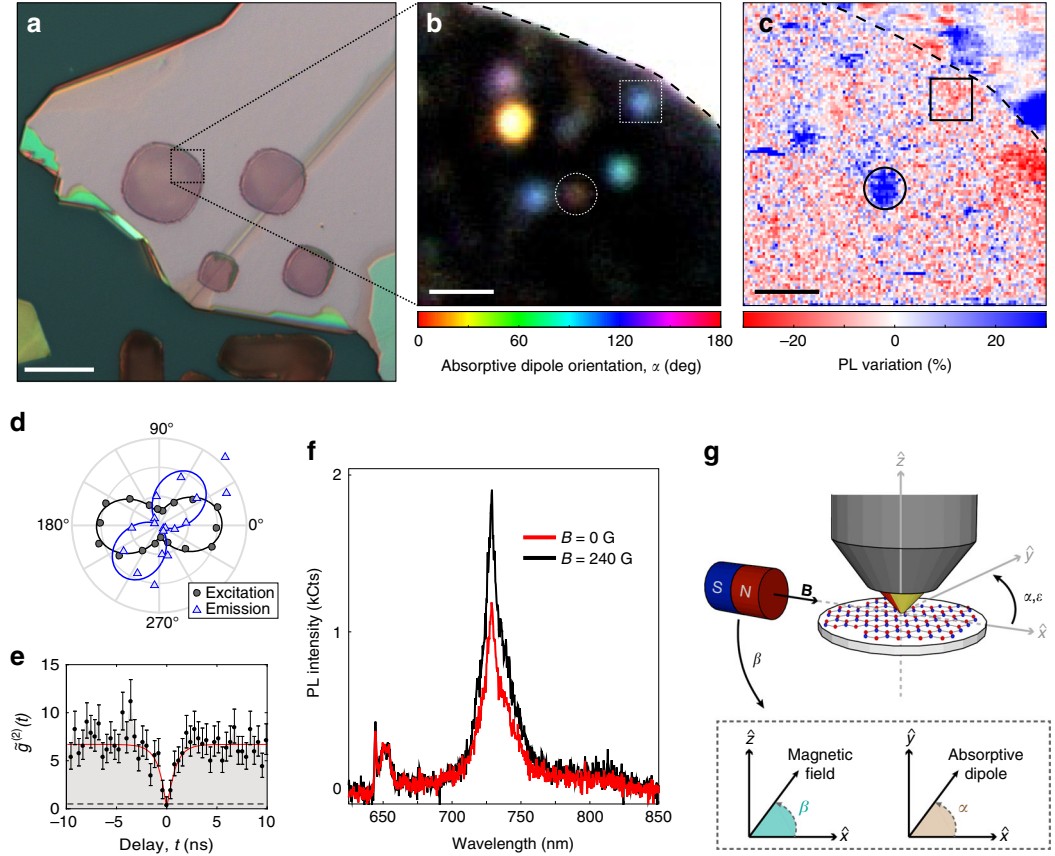

**Fig. 1** Identification of field-dependent quantum emitters. **a** Optical microscope image of an exfoliated h-BN flake on a patterned substrate. Scale bar denotes 10 μm. **b** Polarization-resolved PL image of suspended h-BN [denoted by the dashed box in **a**] at $B = 0$ G. The dashed curve indicates the edge of the suspended region. Color and brightness denote the absorptive dipole orientation and PL intensity, respectively[21]. Scale bar denotes 1 μm. **c** Background-subtracted differential PL variation image from **b** identifying changes due to an in-plane magnetic field ($B = 240$ G). Blue (red) denotes increased (decreased) PL when $B \neq 0$. Scale bar denotes 1 μm. **d–f** correspond to the QE circled in **b** and **c**: **d** Background-subtracted PL excitation (circles) and emission (triangles) polarization dependences. Curves denote fits to the data. **e** Photon autocorrelation function (points) with a fit to a three-level emission model (curve). Data are corrected for a Poissonian background, and error bars represent the Poissonian uncertainty based on photon counts in each bin. The dashed line shows the single-photon emission criterion. **f** PL spectra with and without an in-plane magnetic field parallel to the QE's absorptive dipole. **g** Illustration of the coordinate system for magnetic fields with respect to the microscope objective and sample. $\beta$, in the $\hat{x}$-$\hat{z}$ plane, defines the angle of the magnetic field with respect to the sample plane and $\alpha$ ($\varepsilon$), in the $\hat{x}$-$\hat{y}$ plane, denotes the absorptive (emissive) dipole angle

modulate the steady-state PL. Figure 3a, b shows the observed photon autocorrelation function for several settings of the sample orientation and $B$. Universally, the QE exhibits antibunching at short ($t \lesssim 1$ ns) delay times and bunching over longer ($t \approx 1$ μs) times, qualitatively similar to previous observations of h-BN's QEs[16,17,20,21,35]. We fit the data using an empirical model: $g^{(2)}(t) = 1 - C_1 e^{-t/\tau_1} + \sum_{i=2}^{n} C_i e^{-t/\tau_i}$, where $n = 2$ or 3 depending on the shape of the data; see Section IE in the Supplementary Information. For quantitative comparisons with simulations, we also calculate background-corrected values, $\tilde{C}_i$, as described in the Methods.

The dominant field effect appears in the amplitude of the leading bunching component, $\tilde{C}_2$, which decreases (increases) when the steady-state PL becomes brighter (dimmer) [Fig. 3c, d]. Meanwhile, the bunching timescale remains nearly constant at $\tau_2 \approx 1.4$ μs. This behavior is consistent with a QE model including one or more metastable dark states and an ISC modulated by **B**. In this model, a larger bunching amplitude reflects an increase in the steady-state population trapped in the dark state, and correspondingly lower PL. Interestingly, a third-lifetime component with $\tau_3 \approx 16$ μs is required to capture the autocorrelation shape when **B** = 0 G, but this component vanishes when **B** is in plane [Fig. 3b, inset].

**Modeling spin-dependent optical dynamics.** We use a semi-classical master equation to simulate QE optical dynamics, where the relative transition rates between spin and orbital sublevels are determined by the symmetry-defined Hamiltonian and a set of empirical parameters (see Methods). We consider systems characterized by the point group $C_{2v}$, encompassing many simple defects in multilayer h-BN including vacancy-impurity complexes such as $C_B V_N$ and distorted vacancies such as $N_B V_N$. Using molecular-orbital theory, we consider all possible combinations of three mid-gap, single-particle orbital levels that can encompass an optical ground and excited state with in-plane optical dipole transitions as well as at least one intermediate state from a different spin manifold [Fig. 4a, b].

We further consider configurations with total spin $S \in \left\{ 0, \frac{1}{2}, 1, \frac{3}{2} \right\}$. The lack of symmetry-protected orbital multiplets in $C_{2v}$ makes configurations with $S > \frac{3}{2}$ energetically unfavorable. An applied magnetic field mixes the spin sublevels of configurations with $S \geq 1$ with a pattern determined by the zero-field splitting terms. Crucially, although the spin eigenstates for an $S = 1$ Hamiltonian vary with 180° periodicity as a function of in-plane field orientation, the mixing and ISC spin-selection rules can lead to 90° periodicity in the steady-state PL and autocorrelation parameters [Fig. 4]. On the other hand, for $S = \frac{3}{2}$, the spin

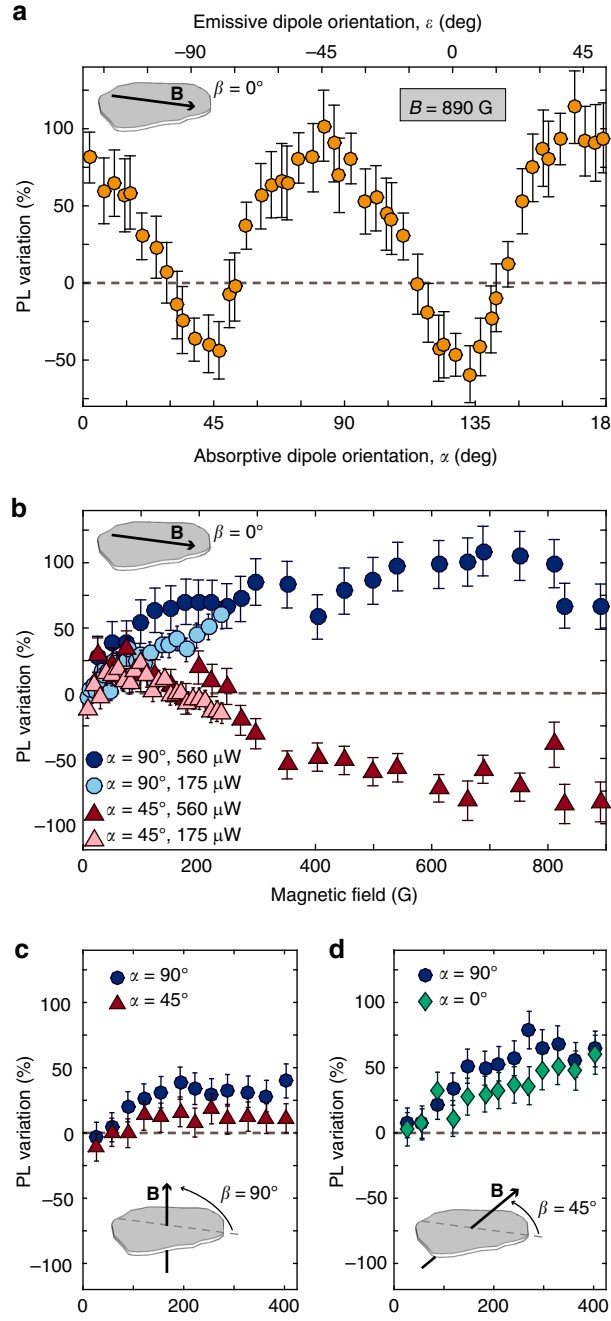

**Fig. 2** Anisotropic magnetic-field-dependent PL. **a** PL variation as a function of the relative orientation between the emitter's optical dipoles and an in-plane $B = 890$ G. **b–d** PL variation for magnetic fields parallel (**b**, $\beta = 0°$), perpendicular (**c**, $\beta = 90°$), and at 45° (**d**, $\beta = 45°$) to the sample plane. All data are taken at 560 μW (measured immediately before the objective) unless otherwise specified. Data in **a** are binned every 3° and at a minimum resolution of 5 G in **b–d**. Bins contain between 1 and 3 measurements, and the weighted mean for each bin is shown. Error bars represent the experimental uncertainty derived from the weighted standard deviation of measurements in each bin (estimated from fits to spatial Gaussian functions or the variance of steady-state PL, as available) along with an overall contribution from the average uncertainty across all field settings

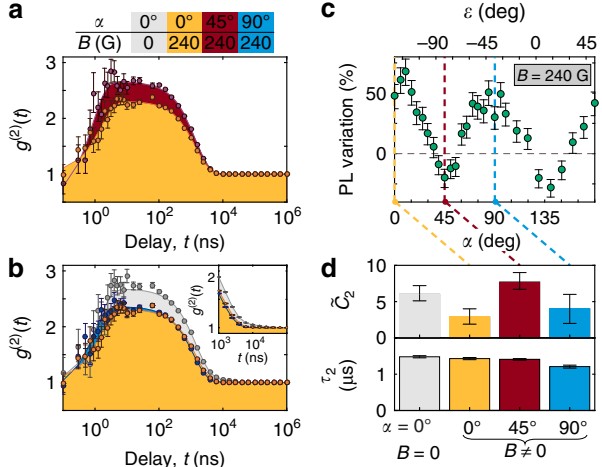

**Fig. 3** Photon emission dynamics. **a, b** Measurements of the photon autocorrelation function for different orientations of the defect with respect to an in-plane magnetic field ($\beta = 0°$) as indicated by the color-coded caption. No background correction is applied to the data, and error bars represent the Poissonian uncertainty based on the photon counts in each bin. **b, inset** Detailed view of the long timescale component ($\tau_3$) visible only when $B = 0$ G. **c** PL variation as a function of sample orientation under an in-plane magnetic field at 240 G. Error bars are determined in the same manner as for Fig. 2a. **d** Best-fit values of the background-corrected bunching amplitude, $\tilde{C}_2$, and corresponding timescale, $\tau_2$, for the data in **a** and **b**. Error bars represent 68% confidence intervals from fits to the data in **a** and **b**, propagated to corresponding background-corrected values including experimental uncertainty in the signal and background

eigenstates are 360° periodic; see Supplementary Figure 8. Furthermore, spin-dependent selection rules do not naturally arise for doublet–quartet transitions in $C_{2v}$, since there is only one double-group representation that must characterize all eigenstates

with half-integer spin. We therefore argue that singlet-triplet configurations are most likely to explain the observed behavior.

Of all the configurations we considered, the level diagrams in Fig. 4a, b most closely match the observations. Both models exhibit 90°-periodic PL variations as a function of in-plane field angle [Fig. 4c, d], with corresponding changes in the intermediate bunching parameters $\tilde{C}_2$ and $\tau_2$ [Fig. 4f, g]. However, simulations of the triplet-ground-state model predict larger variations in $\tau_2$ than we experimentally observe. Moreover, the simulated PL is at a maximum when the triplet optical dipole [gray arrow in Fig. 4c] is aligned or perpendicular to $B$, whereas experimentally we observe a minimum when $\varepsilon = 0°$ or 90° (we assume the emission-dipole axis reflects the QE's underlying symmetry). The singlet-ground-state configuration of Fig. 4b matches the experiments on these points, hence we tentatively identify it as a potential model for the physical system.

## Discussion

So far we have not considered possible chemical structures that could produce the proposed level diagram. Recent calculations[16,30–32] focus on simple configurations with light elements such as $C_BV_N$ or $N_BV_N$. These defects have $C_{2v}$ symmetry and share some features with our models. However, electronic structure calculations in 2D materials remain challenging[31], and uncertainty persists regarding the energy-level ordering even for these simple candidates. Exploration of structures involving heavier elements or larger complexes remains an important goal, ideally guided by atom-scale structural imaging[36] correlated with optical experiments.

While the models in Fig. 4 capture many experimental features, they do not account for all observations. In particular, the simulations in Fig. 4d predict a minimum for the PL at $B = 0$ G,

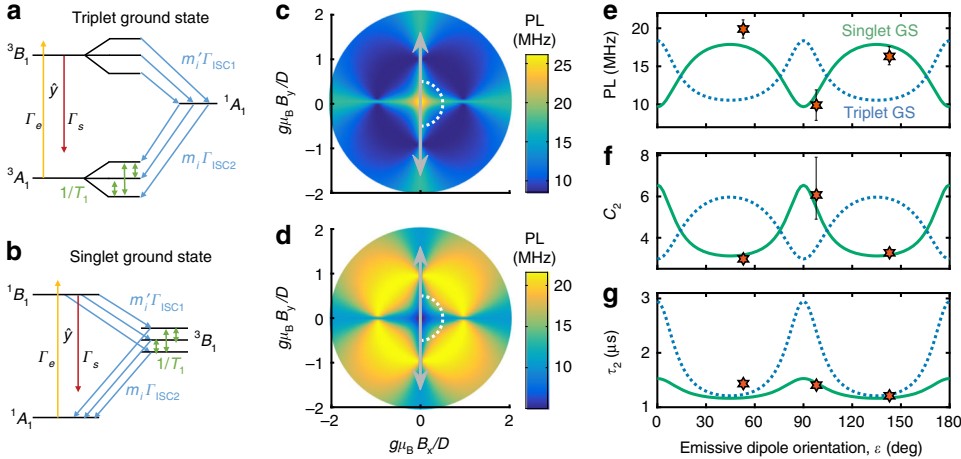

**Fig. 4** Inter-system crossing models. **a, b** Energy-level diagrams showing the symmetry-allowed transitions for a triplet-singlet ISC (**a**) and a singlet-triplet ISC (**b**). **c, d** Steady-state PL simulations for level diagrams in **a** and **b**, respectively, as a function of in-plane magnetic field ($\beta = 0°$). **e, f** Simulated PL amplitude (**e**), bunching amplitude (**f**), and bunching time (**g**) as a function of field orientation, $\varepsilon$, shown as dashed blue and solid green curves for calculations along the dashed curves in **c** and **d**, respectively [$g\mu_B B/D = 0.5$]. Star symbols indicate background-corrected measurements. PL measurements in **e** are multiplied by 1000, and error bars are derived from steady-state PL measurements at comparable field values from Fig. 3c. Error bars in **e** represent uncertainty in spatial Gaussian fits to PL images and steady-state variations for field-dependent measurements. Error bars in **f** and **g** represent 68% confidence intervals from fits to raw autocorrelation data, propagated to corresponding background-corrected values including uncertainty in the signal and background

with no change when $\mathbf{B}||\hat{z}$. Also, whereas longer-lifetime bunching components do emerge from the simulations in certain circumstances, we have been unable to quantitatively reproduce the observations in Fig. 3b using a single set of field-independent parameters. These discrepancies could be related to hyperfine or strain coupling, which is not included in our model, but likely becomes important near $B = 0$ G; see Section IIB of the Supplementary Information.

Future experiments are required to answer these important questions. Field-dependent emission appears to be relatively rare for h-BN's visible emitters, occurring for only a few percent of spots in our samples, but the underlying difference between field-dependent and field-independent emitters remains unknown. Low-temperature optical spectroscopy may aid in identifying optical features specific to QEs which exhibit field-dependent emission. Other measurement modalities, especially optically detected magnetic resonance (ODMR), will be crucial to confirm the predictions of our model and to determine the underlying spin Hamiltonian parameters. Calculations suggest a large ODMR contrast will be observed under the right conditions; see Section IIC of the Supplementary Information. From a materials perspective, significant further work is needed to reproducibly create these spin defects and incorporate them in devices.

The observation of dramatic, room-temperature magnetic-field-induced modulation of single-photon emission in h-BN expands the material's role for use in quantum science and technology. Nanophotonic and nanomechanical devices will exploit optically addressable spins in h-BN for quantum optics[2,15,37] and optomechanics[27,38]. QE electron spins can be used as actuators to address nearby nuclear spins[13,39], offering a platform to study strongly interacting spin lattices[40] and perform quantum simulations[41]. As sensors, the striking PL variation in response to relatively weak magnetic fields bodes well for ultrasensitive detection of nanomagnetism[42,43] and chemical characterization[5,6]. A spinless singlet-ground-state, as proposed in our electronic model, benefits these applications by removing electron-induced nuclear decoherence and unwanted sensor backaction[44].

Additionally, van der Waals heterostructures offer opportunities to engineer the QEs' local environment and control their

functionality, enabling alternative mechanisms for electro-optical addressing. For example, QE spins in h-BN could couple to free carriers or excitons in graphene or transition-metal dichalcogenides, where spin-dependent quantum emission in h-BN could be used to initialize or read out spin-valley qubits for cascaded information transfer between layers[45,46].

## Methods

**Sample preparation and mounting.** Earlier work highlighted the strong influence of substrate interactions during irradiation and annealing treatments on h-BN's visible fluorescence[21]. To eliminate these effects, we study emitters present in freely-suspended h-BN membranes that have been exfoliated from commercially available bulk single crystals and treated as described. All measurements are performed in ambient conditions.

H-BN samples are prepared by exfoliating single-crystal h-BN purchased from HQ graphene onto patterned 90 nm-thick thermal SiO$_2$ on Si substrates. Patterned holes several microns in diameter are created in the support wafer by optical lithography followed by dry etching, creating etched holes several microns wide and ~5 μm deep in the substrate. Exfoliated single-crystal h-BN flakes of interest are those which suspend over a patterned hole[21]. Film thickness is determined using atomic force microscopy; the flake studied here is ~400 nm thick in the vicinity of the suspended region; see Supplementary Figure 2.

Following exfoliation, samples undergo an O$_2$ plasma clean in an oxygen barrel asher (Anatech SCE 108) and are annealed in Ar at 850 °C for 30 min. They are imaged using a scanning electron microscope (SEM) operating at 3 kV (FEI Strata DB235 FIB SEM), generally for <5 min, though the samples may be in the chamber for as long as 30 min. Following the SEM, the samples are again annealed in Ar at 850 °C for 30 min. This process reliably creates individually addressable quantum emitters in suspended regions of the h-BN flakes, of which a few percent show a magnetic-field-dependent response.

An exfoliated and prepared sample is mounted on a rotation stage enabling in-plane rotation of the sample, and thus the optical dipoles of individual defects, with respect to the rest of the setup in a home-built confocal fluorescence microscope with 592 nm continuous wave excitation; see Supplementary Figure 1. Excitation powers used for both bleaching and imaging range from 175 to 550 μW measured immediately before the objective and the PL variation is roughly constant over this range; see Supplementary Figure 3. Additional control over the direction of excitation linear polarization is facilitated with a half-waveplate. An external magnetic field is applied using neodymium magnets mounted on a home-built goniometer that enables variations between the direction of the applied field and the sample plane, as shown in Fig. 1g. Changing the distance between the magnet and the sample allows for a range of applied magnetic fields from 0 to 890 G.

**PL images.** Composite polarized PL images as in Fig. 1b are constructed from a series of confocal PL scans recorded for 4 different linear polarization settings of

the 592-nm excitation laser (0°, 45°, 90°, and 135°), while collecting PL between 650 and 900 nm. The individual images are colorized according to the polarization setting, registered to one another, and summed to create a composite image such that the resulting color, value, and saturation correspond to excitation dipole orientation, visibility, and PL brightness, respectively[21,47].

Differential PL images are as in Fig. 1c are constructed using the value (i.e., brightness) coordinate from composite images acquired with and without a magnetic field. A small constant PL variation [≈8% in Fig. 1c], calculated by averaging over all pixels, is subtracted to account for field-induced changes to the microscope's alignment.

**PL spectra.** PL spectra are taken using a Princeton Instruments IsoPlane 160 spectrometer and a PIXIS 100 CCD with a spectral resolution of 0.7 nm. Multiple exposures (>2) are collected, dark count subtracted and cosmic ray rejected, then averaged together. PL spectra are not corrected for wavelength-dependent photon collection efficiencies. A 633 nm long pass edge filter (Semrock, BLP01-633R-25) in the collection line blocks the 592 nm laser. In Fig. 1f, the h-BN Raman line is visible at the edge of the collection band, at ~644 nm. The field-independent feature around 650 nm is associated with the background.

**PL variation with magnetic field.** Eliminating experimental artifacts that can indicate a false response to magnetic fields is key to determining the field-dependence of QEs in h-BN. We took steps to account for experimental artifacts including plotting the percent variation in the PL rather than the raw PL with and without a magnetic field. To calculate the PL variation at different ($\alpha$, $\varepsilon$) orientations and at different magnetic field strengths, the background-subtracted PL is determined from a combination of Gaussian fits to PL images and, where available, measurements of the time-averaged emission rate detected by focusing directly on the emitter for 30–120 s at each setting. All emission-rate data is background subtracted before forming the PL variation ratio. The background determined via Gaussian fits to confocal PL images is used to background subtract the time-averaged emission-rate data taken under identical conditions. At B = 0 G, the orientation-dependent transmission of PL from the circled emitter through the collection line of the confocal microscope is measured in order to normalize for the small variations (<6%) in PL that occur when the sample is rotated. At each sample orientation, the excitation polarization is aligned with the absorptive dipole.

**Autocorrelation analysis.** Autocorrelation data is obtained using a Hanbury Brown-Twiss setup with a time correlated single-photon counting module (Pico-Quant PicoHarp 300) in time-tagged, time-resolved collection mode. The second-order autocorrelation function, $g^{(2)}(t)$, is calculated from the photon arrival times using the method described in ref. [48], and the curves are fit using empirical functions as described in the text. The choice of model is based on the quality of weighted least-squares fits accounting for the Poissonian uncertainty of each bin.

Background-corrected amplitudes, $\tilde{C}_i$, are calculated using separate calibration measurements of the signal-to-background ratio at each field setting. Using an average measurement of the Poissonian PL background taken from several nearby locations on the suspended membrane, we estimate the background-correction parameter, $\rho = I/(I + I_{bkgd})$, where $I$ is the QE PL and $I_{bkgd}$ is the background PL. The background-corrected amplitudes are then given by $\tilde{C}_i = C_i/\rho^2$. Best-fit parameters and their background-corrected values for all autocorrelation measurements are listed in Supplementary Table 1. Confidence intervals reflect uncertainty in the fits and the measurement of $\rho$.

The short-delay autocorrelation data in Fig. 1e are background corrected in a similar manner using the following relation[49]:

$$\tilde{g}^{(2)}(t) = \frac{g^{(2)}(t) - (1 - \rho^2)}{\rho^2}. \tag{1}$$

The underlying data are the same as in Fig. 3a (red points), rebinned over a linear scale. Since the range of delays is much smaller than the shortest bunching timescale, we fit these data using a simplified empirical function,

$$\tilde{g}^{(2)}(t) = 1 - \tilde{C}_1 e^{-|t|/\tau_1} + \tilde{C}_2, \tag{2}$$

from which we determine a best-fit value $\tilde{g}^{(2)}(0) = 1 - \tilde{C}_1 + \tilde{C}_2 = -0.2 \pm 0.9$, satisfying the single-emitter threshold, $\tilde{g}^{(2)}(0) < 0.5$, by $0.8\sigma$. The accuracy of this measurement is limited by shot noise and detector timing jitter due to the short antibunching timescale, $\tau_1 = 0.8 \pm 0.2$ ns.

**Molecular-orbital theory and optical dynamics simulations.** The goal of our theoretical study is to enumerate a set of simplified models for defect electronic structure based on symmetry considerations[50], and then to perform semiclassical calculations to simulate their optical and spin dynamics under steady-state illumination, for comparisons with experimental results. To that end, we do not start with a particular defect model and study it in detail; rather we explore the qualitative similarities and differences between various electronic configurations in an effort to narrow the space of possibilities. We hope this will motivate future efforts

to compare these qualitative predictions with quantitative, ab initio, calculations of prospective defect configurations in h-BN.

The starting point for any calculation in molecular-orbital theory is the identification of the relevant point group describing the symmetry of the molecule or defect system. We focus on the point group $C_{2v}$, for reasons described in the text, and we choose a coordinate system with $\hat{x}$ as the principal symmetry axis, lying in the h-BN plane, with the $\hat{z}$-axis oriented normal to the h-BN plane. Further details including the character table and group multiplication table for $C_{2v}$ are available in the Supplementary Methods (Section IG).

Here, we consider only the electronic degrees of freedom, i.e., neglecting hyperfine coupling with nuclear spins. (see the Supplementary Discussion Section IIB for a discussion on the role of hyperfine interactions.) In $C_{2v}$ symmetry, the Hamiltonian for an electronic configuration with total spin S takes the form:

$$\hat{H} = g\mu_B \mathbf{B} \cdot \hat{\mathbf{S}} + D\left(\hat{S}_x^2 - \frac{1}{3}S(S+1)\right) + E(\hat{S}_y^2 - \hat{S}_z^2), \tag{3}$$

where $\mu_B$ is the Bohr magneton and $\hat{\mathbf{S}}$ is the spin projection operator. Because spin–orbit coupling in h-BN is relatively weak, we assume that the components of the g-tensor are isotropic and nearly equal to the bare value, g ~ 2. The parameters D and E are zero-field splitting parameters; both terms are nonzero in $C_{2v}$, in contrast to higher-symmetry cases such as $C_{3v}$ or $D_{3h}$, where E vanishes due to symmetry. Their origin can be either first-order spin–spin or second-order spin–orbit interactions, although spin–spin interactions are likely to dominate due to the weak spin–orbit coupling in h-BN. For specific orbital configurations their values can be calculated explicitly in terms of two-electron integrals[32,51,52], but in our general treatment D and E are empirical parameters. Explicit matrix representations of Eq. (3) are provided in Section IG of the Supplementary Methods.

For the remainder we focus on the case of singlet-triplet configurations, since they are more consistent with our experimental observations than doublet–quartet configurations based on symmetry arguments (see Section IG 2 of the Supplementary Methods). Beginning with a minimal configuration of arbitrary single-particle orbitals (two for the case of a singlet-ground state, three for a triplet-ground state), we construct Jablonski diagrams including optical ground and excited states together with possible metastable states with different S. A consideration of the allowed optical dipole selection rules reduces the number of configurations, since in experiments we observe linearly polarized, in-plane emission. Nonradiative ISC transitions are assumed to arise from spin–orbit interactions. The selection rules are determined by identifying symmetrized spin–triplet basis states, $\{|s_x\rangle, |s_y\rangle, |s_z\rangle\} \sim \{A_2, B_2, B_1\}$, and using the group multiplication table to determine the spin state (if any) whose spin–orbit symmetry matches that of an available spin–singlet configuration.

We simulate the orbital and spin dynamics of these systems under optical illumination using a semiclassical master equation model to capture the changing population of individual states due to the coupling rates in a particular electronic level structure [e.g., Fig. 4a, b]. The ISC spin-selection rules are encapsulated in the coupling coefficients,

$$m_i = \sum_\mu p_\mu |\langle s_\mu | s_i \rangle|^2, \tag{4}$$

where $\mu \in \{x, y, z\}$ and $i \in \{1, 2, 3\}$. Here, $|s_i\rangle$ are the field-dependent spin eigenstates determined from Eq. (3), and $p_\mu$ are the normalized selection rules between the singlet and the corresponding symmetrized basis states $|s_\mu\rangle$. This formulation corresponds to the usual assumption that the ISC transitions are incoherent, i.e., that the triplet state resulting from an ISC is described by a density matrix

$$\rho_{triplet} = m_1 |s_1\rangle\langle s_1| + m_2 |s_2\rangle\langle s_2| + m_3 |s_3\rangle\langle s_3|. \tag{5}$$

We further include spin relaxation through a set of uniform transition elements connecting all three pairs of triplet states, at the rate $1/T_1$.

Ultimately, the master equation takes the form $\dot{\mathbf{x}} = R\mathbf{x}$, where R is the rate matrix for the electronic structure in question, and $\mathbf{x}$ is a vector of corresponding state populations. The steady-state population is calculated from the solution of $\langle\dot{\mathbf{x}}\rangle = R\langle\mathbf{x}\rangle = 0$, i.e., from the null space of R. Assuming the radiative transitions are spin conserving, the steady-state PL is given by $\langle PL\rangle = \Gamma_s \sum_{i \in ES}\langle x_i\rangle$, where GS (ES) refers to the ground state (excited state). The autocorrelation function is calculated by numerically integrating the master equation starting with an initial condition corresponding to the configuration that follows emission of a photon. [Note that, in general $\mathbf{x}_{GS}(0) \neq \langle\mathbf{x}_{GS}\rangle / \sum\langle\mathbf{x}_{GS}\rangle$]. Given this initial condition, the autocorrelation function is related to the subsequent evolution of the excited-state population via

$$g^{(2)}(t) = \frac{PL(t)}{\langle PL\rangle} = \frac{\sum_{i \in GS, j \in ES} R_{ij} x_j(t)}{\langle PL\rangle}. \tag{6}$$

Additional details regarding the modeling for various spin configurations, energy-level structures, and PL, autocorrelation, and ODMR simulations are presented in the Supplementary Information.

**Code availability**. Codes used for the analysis of autocorrelation data and master-equation simulations of optical dynamics are available from the corresponding author upon request.

## Data availability

The data that support the findings of this study are available from the corresponding author upon reasonable request.

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

## Acknowledgments

This work was supported by the Army Research Office (W911NF-15-1-0589). M.W.D. was supported by the Australian Research Council (DE170100169). The authors gratefully acknowledge use of facilities and instrumentation supported by the NSF through the University of Pennsylvania Materials Research Science and Engineering Center (MRSEC) (DMR-1720530). This work was carried out in part at the Singh Center for Nanotechnology, which is supported by the NSF National Nanotechnology Coordinated Infrastructure Program (NNCI-1542153). The authors thank Jennifer Saouaf and

Richard Grote for assistance in sample preparation, Tzu-Yung Huang for assistance with measurements, and Audrius Alkauskas for engaging theoretical discussions.

## Author contributions

A.L.E. and L.C.B. conceived and designed the experiments. A.L.E., D.A.H., R.N.P., and L. C.B. performed the measurements and analysis. A.L.E., M.W.D., and L.C.B. developed the theory and simulations. All authors contributed to writing the manuscript.

## Additional information

**Competing interests:** The authors declare no competing interests.

