## [Peer Review File · Nature Communications]

Reviewers' comments:

Reviewer #1 (Remarks to the Author):

Exarhos et al. submit a paper entitled "Spin-dependent quantum emission in hexagonal boron nitride at room temperature" for publication in Nature Communications. This work is an investigation of spin-related effects in the emission of point defects in hexagonal boron nitride (h-BN), a feature that is still missing in this material, although many studies have reported bright single photon sources in h-BN. The objective of the present submission is clear and timely, and frankly speaking, I was quite excited when starting to read this paper. However, evidence to support this claim is still lacking, for the following reasons:

- the authors have only observed a magnetic field-dependent PL signal (not a spin-dependent effect)
- this observation is made for only a few percent of spots in the sample
- the emitters are located in patterned areas, of an exfoliated h-BN flake suspended on a patterned substrate
- last but not least, no ODMR is reported

Therefore, we are far from a systematic, robust and direct signature for spin-dependent quantum emission in h-BN. As written in the abstract, "field-dependent variations in the steady-state PL and photon emission statistics are CONSISTENT [but no more] with an electronic model featuring a spin-dependent inter-system crossing". In the absence of ODMR, one could buy the claim for spin-dependent effects if the observation was systematic and independent from the sample preparation/geometry. This is unfortunately not yet the case, and at that stage, I would suggest to the authors to modify the title to "Magnetic field-dependent quantum emission..." and submit this still interesting contribution to a more specialized journal.

Reviewer #2 (Remarks to the Author):

Exarhos et al. studied some photoluminescence (PL) centres and they observed magnetic field dependence in the intensity of the steady-state PL spectrum. They analysed the spectrum and the observed photo-dynamics by constructing a spin Hamiltonian and group theory.

In my opinion, the results of this study do not meet with the high expectations of Nature Communications, both in terms of the format and the achieved results. This paper may suit Physical Review B or related journal.

Let me start with the format of the publication. The Supporting Information is 30 pages long, and in particular, it contains an entire paper about the theoretical approaches, analysis and results (~26 pages long). This habit is controversial: the supporting information is originally for providing extra data to justify the statements in the main text, and it is not intended to write an entire paper which follows or precedes the main text of the paper. The big disadvantage of this habit in this particular case is that the theory "paper" in the Supporting Information is hidden and the theory work would not be granted. The unification of the experimental and theoretical parts into a single paper would greatly enhance the visibility and contribution of the theoretical part to the entire study.

I would like to now briefly study the impact of the paper. Although, the observation of spin-dependent steady-state PL of a quantum emitter in h-BN is important but neither the spin states involved in the PL process nor the atomic configurations of these quantum emitters have been conclusively identified. In my understanding, the conditions of formation of these quantum emitters in h-BN were neither understood, so there is no recipe to produce these quantum emitters by demand. I recognize the great effort toward reaching these goals, however, these efforts were only partially successful. Theoretical considerations tentatively assign these emitters to bearing singlet ground state with spin-1 metastable state. This association is still based on incomplete correspondence between the simulation results and measured data points. I refer to the followings:

Page 2: "Interestingly, a third lifetime component with $\tau_3=16$ microsec is required to capture the autocorrelation shape when $B = 0$ G, but this component vanishes when B is in plane [Fig. 3(b, inset)]."

Does the model explain this behaviour? How the model should be set to account for this observation?

Page 5, Figure 4 (e)-(g): the observed data points are scarce (stars on the plot). The simulation results and discussion would have been much more conclusive after measuring more data points (more angles) for the PL, C2, and τ_2 .

Reviewer #3 (Remarks to the Author):

This paper presents observations of photoluminescence (PL) from defects in h-BN that is dependent on an external magnetic field B . This dependence is characterised with rigorous control of the applied B direction and magnitude, and optical polarization. Further, this behaviour, including bunching characteristics of time binned photons, is shown to be consistent with a defect that contains a spin triplet manifold and singlet ground state. This work is the first that has shown any type of B -field dependence of defects hosted in h-BN. Though this is very preliminary work in the context of spin-photon interfaces, it will be of interest and provide additional motivation to the 2D materials quantum optics community especially with much of the work on defects in h-BN and other 2D materials still at the very exploratory stage. Finding a spin-photon in such materials is one of the major goals in the community.

The work is convincing but I have a few comments below that should be addressed:

- 1) Be consistent on where excitation powers are measured. Preferably state the power after the objective. See fig 2 caption etc...
- 2) Though background subtraction is mentioned for g_2 measurements, when calculating PL variation $((I_B - I_0)/I_0)$ do you subtract the background? I assume you do since the ratio is highly dependent on background. Explain somewhere in the text how this background is subtracted and if spectrometer or APD counts are used.
- 3) In Fig 2a) at $B=890$ G the PL var. is -50% at $\alpha=45$ degs, $\beta=0$ degs. But in fig 2b) the red triangles for same orientation the PL drops close to -100%. Why?
- 4) In last paragraph of section "Variations in steady-state PL" you state Fig 2d suggests an underlying 180 deg symmetry. Not clear to me why this is. Maybe a bit more explanation would help.
- 5) In fig 2b you show data for $\alpha=45$ and 90 degs. Is 0 degs behavior same as 90? If so could state this in the main text.
- 6) In fig 4 caption the description e) is described twice and there is no description for g)
- 7) When referring to background corrected... in the main text for the first time could you state that details are given in methods section.
- 8) In part that reads "Also, whereas longer-lifetime bunching components do emerge from the simulations in certain circumstances, we have been unable to quantitatively reproduce the observations in Fig. 3(c)...". Do you really mean fig 3c)? Not clear how data in that data corresponds to the lifetimes of bunching.
- 9) Could the discrepancies at $B=0$ G field arise from strain, this would mix the triplet state spins? That was the first thing that popped into my mind before hyperfine interactions.

If these are addressed, I would be happy to recommend this for publication.

Itemized responses to the June 25th reviewers' comments concerning our manuscript appear below. Referee comments are formatted in italic font. Responses to each comment are formatted in normal font along with a summary of the action taken in the revised manuscript.

Reviewer #1 (Remarks to the Author):

Exarhos et al. submit a paper entitled "Spin-dependent quantum emission in hexagonal boron nitride at room temperature" for publication in Nature Communications. This work is an investigation of spin-related effects in the emission of point defects in hexagonal boron nitride (h-BN), a feature that is still missing in this material, although many studies have reported bright single photon sources in h-BN. The objective of the present submission is clear and timely, and frankly speaking, I was quite excited when starting to read this paper. However, evidence to support this claim is still lacking, for the following reasons:

- the authors have only observed a magnetic field-dependent PL signal (not a spin-dependent effect)*
- this observation is made for only a few percent of spots in the sample*
- the emitters are located in patterned areas, of an exfoliated h-BN flake suspended on a patterned substrate*
- last but not least, no ODMR is reported*

Therefore, we are far from a systematic, robust and direct signature for spin-dependent quantum emission in h-BN. As written in the abstract, "field-dependent variations in the steady-state PL and photon emission statistics are CONSISTENT [but no more] with an electronic model featuring a spin-dependent inter-system crossing". In the absence of ODMR, one could buy the claim for spin-dependent effects if the observation was systematic and independent from the sample preparation/geometry. This is unfortunately not yet the case, and at that stage, I would suggest to the authors to modify the title to "Magnetic field-dependent quantum emission..." and submit this still interesting contribution to a more specialized journal.

Our Response: We appreciate the reviewer's point of view, and we share their frustration with the diverse and dissimilar experimental characteristics of quantum emitters in h-BN. The reviewer is absolutely correct that we are far from a systematic understanding of spin-dependent emission from h-BN, or, for that matter, a firm identification of the defects responsible for the emission. Significant additional work, far beyond the scope of this manuscript, is required to establish concrete links between physical defect configurations, their electronic structure, optical and spin dynamics, and the effects of external perturbations. Eventually, we hope that future experimental and theoretical efforts to understand the detailed electronic structure and optical dynamics that give rise to field-dependent fluorescence, supported by steady improvements in the purity of starting material and reliable methods for defect creation in h-BN will address the reviewer's concerns regarding the ability to create and isolate emitters with uniform properties. For now, the heterogeneity of quantum emitter observations in h-BN makes our task harder, since we cannot probe ensembles of defects that all exhibit similar characteristics, as has usually been essential for understanding defects in other materials.

Against this backdrop, we feel that our work represents a significant contribution to the conversation about quantum emitters in h-BN. Our experiments unequivocally demonstrate a striking, room-temperature, magnetic-field-induced modulation of quantum emission, similar to optically addressable spin defects like the nitrogen-vacancy center in diamond. We have observed the effect for multiple emitters in different samples. The fact that field-dependent effects appear for only a fraction of emitters is consistent with the broader heterogeneity of emission characteristics in this material (e.g., spectral,

brightness, and polarization properties), supporting the interpretation that multiple defect configurations or distinct species could be involved.

Our intention in studying quantum emitters in suspended areas of exfoliated, single-crystal h-BN flakes was to remove some of this uncertainty, isolating the source of the quantum emission to the h-BN itself and eliminating the possibility of emitters existing in the substrate or at an interface with the h-BN. We note that many emitters do exist in the supported regions of the exfoliated films and some appear to show a magnetic-field-dependent modulation, but in our samples the emitters in supported regions are generally not sufficiently isolated for individual study.

We appreciate the reviewer's comment that "spin-dependent" and "magnetic-field-dependent" are synonymous. We remain confident of the connection to spin, based on a consideration of the energy scales involved (e.g., the range of magnetic field strength over which modulation occurs) and the broad qualitative agreement of our spin-dependent optical dynamics models with the observed photon emission statistics. These considerations rule out other possibilities such as orbital magnetic perturbations. However, we have accepted the reviewer's suggestion to change the title to emphasize that we are observing a magnetic-field-dependent effect.

Finally, we agree with the reviewer that ODMR experiments will be an important next step to verify our interpretation of a spin-dependent inter-system crossing transition. We are working towards such experiments but so far do not have any results to present. In the absence of bulk EPR signatures from defect ensembles, we do not know what frequencies are required. Furthermore, if the spin exists in a metastable state (as our model predicts) or with a short lifetime due to strong hyperfine or other relaxation mechanisms, it may be necessary to apply strong ac driving fields in order to observe ODMR, and this will require customized sample geometries with on-chip antennas. These experiments are well outside the scope of the current manuscript, but we have included predictions for the ODMR response when various spin transitions are driven in the supporting information to guide future experiments (Fig S19). Ultimately, we believe the sum of our observations – especially the anisotropic modulations of steady-state PL due to applied dc magnetic fields, coincident with changes in the optical emission dynamics – are sufficient to support our interpretation for a spin-dependent optical response.

Action Taken: The title of the manuscript has been changed to "Magnetic-Field-Dependent Quantum Emission in Hexagonal Boron Nitride at Room Temperature." To address the issue of reliable creation of field-dependent emitters, we added the following sentence to the second paragraph of the methods discussing the sample treatment process:

"This process reliably creates individually addressable quantum emitters in suspended regions of the h-BN flakes, of which a few percent show a magnetic-field-dependent response."

Reviewer #2 (Remarks to the Author):

Exarhos et al. studied some photoluminescence (PL) centres and they observed magnetic field dependence in the intensity of the steady-state PL spectrum. They analysed the spectrum and the observed photo-dynamics by constructing a spin Hamiltonian and group theory.

In my opinion, the results of this study do not meet with the high expectations of Nature Communications, both in terms of the format and the achieved results. This paper may suit Physical Review B or related journal.

Let me start with the format of the publication. The Supporting Information is 30 pages long, and in

particular, it contains an entire paper about the theoretical approaches, analysis and results (~26 pages long). This habit is controversial: the supporting information is originally for providing extra data to justify the statements in the main text, and it is not intended to write an entire paper which follows or precedes the main text of the paper. The big disadvantage of this habit in this particular case is that the theory “paper” in the Supporting Information is hidden and the theory work would not be granted. The unification of the experimental and theoretical parts into a single paper would greatly enhance the visibility and contribution of the theoretical part to the entire study.

Our Response: We understand the reviewer’s point here and sympathize with their personal views. However, we want to emphasize that this is primarily an experimental work reporting a new and surprising observation of magnetic-field-dependent quantum emission in h-BN, the implications of which indicate that the realization of spin-based quantum technologies is possible in h-BN. We believe the current format of the paper correctly keeps the emphasis on the experimental data, demonstrating the presence of magnetic-field-dependent quantum emitters in h-BN. Qualitatively, the observation of anisotropic modulation of emission dynamics as a function of a dc magnetic field is readily interpreted in the context of a spin-dependent inter-system crossing, since this is the mechanism that produces similar effects in other solid-state defects (*e.g.* the NV center in diamond). Our theory and simulations support this interpretation and help to make the comparison more quantitative, explaining some (but not all – see further comments below) of the features in the experiment and helping to constrain the space of likely electronic-structure models. The approach represents a mostly straightforward application of well-established molecular-orbital-theory methods and master-equation simulations to the specific problem of C_{2v} -symmetric defects in h-BN as referenced in the Supporting Information. We have tried to describe our approach as thoroughly as possible in the Supporting Information in order to aid other researchers in replicating our models and approach as applied to h-BN or other new defect systems. We appreciate that the reviewer considers this description a paper in its own right!

In order to address this point by the reviewer, and in light of the flexible formatting requirements of *Nature Communications*, we have included a broad discussion of the theoretical approach in the Methods section of the main paper. The remainder of the detailed presentation in the Supporting Information is referenced at several locations throughout the text. We certainly do not intend for this material to be “hidden,” but we also hesitate to place undue emphasis on the details of simulations for particular electronic configurations, etc., given the preliminary stages of experiments in this materials platform. Hopefully this compromise will preserve the spirit of the paper while also providing useful supporting information for other researchers to replicate and expand upon our results.

Action Taken: An overview of the theoretical methods and simulation models now appears in a new subsection of the Methods: “Molecular Orbital Theory and Optical Dynamics Simulations”

I would like to now briefly study the impact of the paper. Although, the observation of spin-dependent steady-state PL of a quantum emitter in h-BN is important but neither the spin states involved in the PL process nor the atomic configurations of these quantum emitters have been conclusively identified. In my understanding, the conditions of formation of these quantum emitters in h-BN were neither understood, so there is no recipe to produce these quantum emitters by demand. I recognize the great effort toward reaching these goals, however, these efforts were only partially successful.

Our Response: We again acknowledge and sympathize with the reviewer’s point of view. We refer the reviewer to our response to the related comments by Reviewer #1 regarding the nascent stage of research with quantum emitters in h-BN and the incomplete understanding of the defects’ chemical & electronic

structures. We believe, however, that this work plays a key role in furthering this understanding. Our observation that h-BN hosts a previously unreported field-dependent emitter species will encourage additional research toward clarifying their electronic and chemical structures as well as their potential applications for spin-based quantum technologies. We ask the reviewer to reconsider his or her evaluation of the impact of our results in this context.

Theoretical considerations tentatively assign these emitters to bearing singlet ground state with spin-1 metastable state. This association is still based on incomplete correspondence between the simulation results and measured data points. I refer to the followings:

Page 2: “Interestingly, a third lifetime component with $\tau_3=16$ microsec is required to capture the autocorrelation shape when $B = 0$ G, but this component vanishes when B is in plane [Fig. 3(b, inset)].” Does the model explain this behaviour? How the model should be set to account for this observation?

Our Response: The model does not fully explain the presence of the third lifetime component. (We addressed this in the text, but incorrectly referenced the figure, so this point was unclear.) It has been corrected in the second paragraph of p5:

“Also, whereas longer-lifetime bunching components do emerge from the simulations in certain circumstances, we have been unable to quantitatively reproduce the observations in Fig. 3(b).”

The role of modelling in this manuscript is not to definitively establish that the quantum emitter in question has a specific electronic structure, but rather to consider some of the simplest electronic models to see which could be consistent with the observations. As the reviewer points out, none of the simple models we consider reproduces every feature of the experiments, including the “best match” model of a singlet-ground-state, metastable-triplet configuration. It is possible that more sophisticated models including additional parameters (such as hyperfine coupling, spin-spin coupling, or strain, for example) could account for these observations, and we provide some qualitative arguments in support of this view in the Supporting Information. We believe it would be premature, however, and beyond the scope of this manuscript, to include these factors in the simulations, since it will only introduce additional degrees of freedom. Future experiments and *ab initio* theory are needed to constrain these parameters, including measurements on many more field-dependent defects to understand the statistical variations due to different local environments.

Nevertheless, the fact that magnetic-field-dependent emission can exist at all in h-BN has been an open question and is one that we conclusively answer in the affirmative here.

Page 5, Figure 4 (e)-(g): the observed data points are scarce (stars on the plot). The simulation results and discussion would have been much more conclusive after measuring more data points (more angles) for the PL, C2, and τ_2 .

Our Response: We agree with the reviewer that additional autocorrelation measurements would improve Figure 4. Unfortunately, more extensive data collection of the optical dynamics of the quantum emitter under study here were not possible. The autocorrelation measurement requires significant exposure time of the quantum emitter to the excitation laser at a high power, and after time (many weeks, in this case) this exposure can lead to the quantum emitter disappearing through lattice heating or chemical adsorption, for example. We were attempting to take additional dynamical data at other angles when the quantum emitter disappeared from the sample. Nonetheless, the measurements we acquired help to distinguish between two related electronic-structure models, and we believe this is a compelling part of the story that is essential to include.

Reviewer #3 (Remarks to the Author):

This paper presents observations of photoluminescence (PL) from defects in h-BN that is dependent on an external magnetic field B. This dependence is characterised with rigorous control of the applied B direction and magnitude, and optical polarization. Further, this behaviour, including bunching characteristics of time binned photons, is shown to be consistent with a defect that contains a spin triplet manifold and singlet ground state. This work is the first that has shown any type of B-field dependence of defects hosted in h-BN. Though this is very preliminary work in the context of spin-photon interfaces, it will be of interest and provide additional motivation to the 2D materials quantum optics community especially with much of the work on defects in h-BN and other 2D materials still at the very exploratory stage. Finding a spin-photon in such materials is one of the major goals in the community.

Our Response: We appreciate the reviewer's careful reading of our manuscript and positive comments regarding our work.

The work is convincing but I have a few comments below that should be addressed:

1) Be consistent on where excitation powers are measured. Preferably state the power after the objective. See fig 2 caption etc...

Our Response: Excitation powers were measured immediately before the objective (the objective was removed and a power meter put in its place for the excitation power measurements). We have modified the text in the caption of Fig. 2 and the methods section (sample preparation and mounting) to reflect this.

Action Taken: Added the phrase "measured immediately before the objective" to the Fig. 2 caption and the third paragraph of the Methods section.

2) Though background subtraction is mentioned for g2 measurements, when calculating PL variation $((I_B - I_0)/I_0)$ do you subtract the background? I assume you do since the ratio is highly dependent on background. Explain somewhere in the text how this background is subtracted and if spectrometer or APD counts are used.

Our Response: The data used to calculate the PL variation is background subtracted based on APD counts. We determine the background by performing a 2D Gaussian fit on an APD-acquired PL image of the emitter. The data used for the PL variation plots in Figs. 2a-d, 3a is derived from a combination of the 2D Gaussian fit amplitudes as well as (in the case of Figs. 2b-d) the average of thresholded APD counts over a 30-120 s duration while focusing directly on the emitter. The averaged counts are background subtracted using the background determined from the 2D Gaussian fit to the PL images at the same magnetic field (taken immediately before and after each average emission rate measurement). The combination of these two approaches reduce the effects of slow-timescale blinking and other low-frequency noise in the measurement, however these noise sources are still included in the error bars that indicate the experimental uncertainty of each data point.

Action Taken: We have modified the Methods (*PL Variation with Magnetic Field Measurements*) to include this information:

“All emission rate data is background-subtracted before forming the PL variation ratio. The background determined via the Gaussian fits to the single-photon counter-collected PL images is used to background subtract the time-averaged emission rate data taken under identical conditions.”

3) In Fig 2a) at $B=890G$ the PL var. is -50% at $\alpha=45$ degs, $\beta=0$ degs. But in fig 2b) the red triangles for same orientation the PL drops close to -100%. Why?

Our Response: The data for Fig. 2a and Fig. 2b were taken in two different runs separated by several days. We have attempted to include all sources of experimental uncertainty that we could directly measure (due, e.g. to drift of the optical alignment, low-frequency blinking of the emission, and shot noise due to limited photon acquisitions), however some additional uncertainty might not be reflected in the error bars. The decreased emission data was particularly hard to collect as the emitter becomes very difficult to distinguish from the background at high fields in this orientation, so noise and variation of the emission amplitudes in that situation are unsurprising. Considering especially the scatter in the measurements in Fig 2(b), we do not think there is a statistical inconsistency with the data in Fig 2(a). For example, our model predicts that the emission should be similar at $\alpha = 45$ and 135 deg, and the 135 deg. value in Fig 2(a) is quite similar to the high magnetic field data in Fig. 2(b).

4) In last paragraph of section “Variations in steady-state PL” you state Fig 2d suggests an underlying 180 deg symmetry. Not clear to me why this is. Maybe a bit more explanation would help.

Our Response: We agree that this statement is unclear as it was written. The 180 deg symmetry comment arises as a result of studying both the $\beta=90$ and $\beta=45$ deg behavior at various alphas ($\alpha = 0, 45, 90$). Figs 4(c) and 4(d) show that the PL increases monotonically when $\beta = 45$ or 90 deg, when $\alpha = 0$ or 90 deg. The strength of the PL variation for $\beta = 45$ deg appears to be intermediate between the value when $\beta = 0$ or 90 , which would be consistent with 180 deg symmetry (e.g., a $\cos^2(\beta)$ dependence). This is in striking contrast to the case of different angles α when $\beta=0$. Furthermore, we do observe similar PL variation of the QE when the applied magnetic field is oriented in such a way as to couple the in-plane and out-of-plane emission of the QE at $\beta = 45$, indicating that the out-of-plane component does not influence the in-plane behavior.

Action Taken: We have changed the text as follows:

At the end of the Identification of *Magnetic-Field-Dependent Quantum Emitters* section, we have clarified the dipole and magnetic field orientation angles (changes shown in bold):

“... we explore arbitrary field orientations by rotating the sample about the optical axis, where α (ϵ) denotes the orientation of the absorptive (emissive) dipole **in the plane of the sample (x-y plane)**, relative to x, and by adjusting a magnet goniometer in the x-z plane (**out of the sample plane**), where β is the angle of the field relative to x”

Also, at the end of the *Variations in Steady-State PL* section, the text now reads:

“**This behavior, along with observations of** a similar monotonic increase observed for $\beta = 45^\circ$ [Fig. 2(d)] , suggests an underlying 180° symmetry for rotations about x or y,...”

We have also clarified the magnetic field orientations for Fig. 2 by emphasizing that $\beta = 0^\circ$ in the first sentence of this section.

5) In fig 2b you show data for alpha=45 and 90 degs. Is 0 degs behavior same as 90? If so could state this in the main text.

Our Response: Yes, the 0 deg behavior is expected to be the same as the 90 deg behavior (and 45 and 135 deg are similarly matched). We do not have full data sets at alpha=0 that correspond to the data taken at alpha = 90, but we observed the 90 deg periodicity at multiple field amplitudes (Fig. 2a and Fig. 3c, for example), implying that the 90 deg periodicity persists across different applied magnetic field strengths.

Action Taken: We have modified the main text by adding the following sentence to the first paragraph of *Variations in Steady-State PL*:

“The observed 90° periodicity persists when varying magnetic field strength [Figs. 2(a) and 3(c), for example].”

6) In fig 4 caption the description e) is described twice and there is no description for g)

Our Response: Thank you for pointing this out. We have corrected this in the caption so that (e) refers to the simulated PL amplitude, (f) to the bunching amplitude, C2, and (g) to the bunching time, tau2.

7) When referring to background corrected... in the main text for the first time could you state that details are given in methods section.

Our Response: We have included a reference to the Methods section in the *Identification of Magnetic-Field-Dependent Quantum Emitters* section when describing the autocorrelation function:

“The background-corrected second-order autocorrelation function, $g^{(2)}(t)$, [Fig.1(e)], exhibits an antibunching dip near zero delay that drops below the threshold, $g^{(2)}(0) < 0.5$, indicating the PL is dominated by a single emitter (**see Methods**).”

8) In part that reads “Also, whereas longer-lifetime bunching components do emerge from the simulations in certain circumstances, we have been unable to quantitatively reproduce the observations in Fig. 3(c)...”. Do you really mean fig 3c)? Not clear how data in that data corresponds to the lifetimes of bunching.

Our Response: Thank you for drawing our attention to this. The statement should have referred to Fig. 3(b). It has been corrected.

9) Could the discrepancies at B=0G field arise from strain, this would mix the triplet state spins? That was the first thing that popped into my mind before hyperfine interactions.

Our Response: This is an interesting suggestion. While we have not conducted a full analysis of strain effects, we suspect that strain-induced spin mixing should be relatively weak for C_{2v} -symmetric defects, since there are no symmetry-protected orbital degeneracies to mediate the mixing. However, it is still possible that the local strain is much larger than we anticipate, or accidental degeneracies could play a role. In the Supplemental Information (final paragraph of *Additional Magnetic-Field-Dependent Emitters*), we note that if our model were to take into account “variations in the local defect parameters due, e.g., to strain that shifts of the triplet zero-field splitting terms...” and/or by tuning other rates and

relaxation times, the differences between simulations and experimental data in the autocorrelation functions can likely be reconciled, though given the large number of potential free parameters, we have not explored the effects of strain specifically in this context.

Action Taken: We have therefore noted possible effects of strain explicitly in the main text, adding to the second paragraph on p5:

“These discrepancies [between experimental observations and models] could be related to hyperfine **or strain** coupling, which is not included in our model, but likely becomes important near $B=0$ G.”

We have also addressed the possible effects of strain in the Supporting Information at the end of the Results and Discussion section of the Molecular Orbital Theory section:

“We hypothesize that these discrepancies might be related to other physical effects such as strain - which could lower the defect symmetry and change the dipolar spin-spin interaction of the triplet level. Perhaps more likely, however, are hyperfine or spin-orbit coupling effects between levels that are not currently included in our model, but further experiments will be required to fully answer these and other questions raised by this work.”

If these are addressed, I would be happy to recommend this for publication.

REVIEWERS' COMMENTS:

Reviewer #1 (Remarks to the Author):

I have carefully read the response of the authors and the re-submitted manuscript entitled "Magnetic-field-dependent quantum emission in hexagonal boron nitride at room temperature". I acknowledge the intellectual honesty of the authors, who do not try to oversell their results. To me, this is a very good and interesting paper, but a PRB rather than a Nat. Comm. Similarly to Reviewer #2, I think this is a great (yet incomplete) effort toward reaching defects with useful spin properties in hBN.

Reviewer #2 (Remarks to the Author):

I read the comments of all Reviewers and the reply to all of these comments by the authors, and the revised paper.

Let me state first that I recognize the authors' attempts to carefully consider all the critiques drawn by the Reviewers and improvement in the revised paper by following the guidance from the Reviewers. I appreciate this attitude and the collegial tone. In particular, the paper is now extended with the method from theory in the main paper.

Now I turn to the critical point on the impact of the paper. I recognize that it is difficult to create objective criteria about this issue for a given scientific journal, so thus there is a room for subjective viewpoint or taste of the Reviewers in this regard.

As this is the first report on variation of the PL signal of a single emitter upon external magnetic fields in h-BN, a 2D material, I think that the interpretation of the experimental spectra is highly critical. Direct observation of optically detected magnetic resonance would provide a solid proof that a single photon emitter with net spin has been found (that might be coherently manipulated by optical means). Here, this is not the case. Thus, in my viewpoint, the theory here is required to provide a model that can support that the observed magnetic field dependence of the signal is an inherent property of the selected single photon emitters, and one can fully exclude that the observed features are not related to some "cross-talks" or artefact in the measurements (e.g., the detectors' sensitivity is affected by the presence of the magnet, etc.). That is why I found to be important to demonstrate a convincing correspondence between the theoretical model and the measured data points in Fig. 4 (e)-(g). I understand that it is very challenging to observe new data points (star symbols in the figure). However, novel data coming from extra measurements could strongly support the consistency between the model and the magnetic field dependence of the PL of single photon emitters.

The few single photon emitters that show magnetic field dependent PL are in the class having singlet ground state and a metastable triplet state, presumably with C_{2v} symmetry. Unfortunately, almost all the defects with closed shell singlet electronic configuration have a triplet excited state beside singlet excited states. From optical characterization point of view, PLE measurements (presumably at low temperatures) may reveal special features that could help understanding why these emitters are unique in terms of magnetic field dependent PL compared to the other single photon emitters. I recognize that these measurements would again require additional efforts (that might be not simple and straightforward) from the authors. Nevertheless, that would significantly raise the impact of this study.

All-in-all, in my opinion, this is a very interesting report on spin-dependent PL signal of single photon emitters in h-BN but, at least, the optical (and/or electrical) properties of these single photon emitters should have been more thoroughly studied, in order to raise the impact to the very high scientific standard of Nature Communications. Although, I recognize this is a rigorous

viewpoint that can be considered and evaluated by the Editors of Nature Communications.

Reviewer #3 (Remarks to the Author):

I am happy with the author responses to my queries, with the edited manuscript and to recommend this for publication.

Itemized responses to the November 1st reviewers' comments concerning our manuscript appear below. Referee comments are formatted in italic font. Responses to each comment are formatted in normal font along with a summary of the action taken in the revised manuscript.

Reviewer #1 (Remarks to the Author):

I have carefully read the response of the authors and the re-submitted manuscript entitled "Magnetic-field-dependent quantum emission in hexagonal boron nitride at room temperature". I acknowledge the intellectual honesty of the authors, who do not try to oversell their results. To me, this is a very good and interesting paper, but a PRB rather than a Nat. Comm. Similarly to Reviewer #2, I think this is a great (yet incomplete) effort toward reaching defects with useful spin properties in hBN.

Our Response: We very much appreciate the comments of the reviewer; they have helped us to clarify and improve our manuscript. We understand the opinion of the reviewer, and acknowledge that views regarding novelty and impact are subjective. In our view, this paper warrants readership by the broader audience of *Nature Communications* compared to *PRB* not because it offers the final word on spin-dependent defects in h-BN (far from it!), but because it represents an important discovery that will motivate future work in this area by others.

Reviewer #2 (Remarks to the Author):

I read the comments of all Reviewers and the reply to all of these comments by the authors, and the revised paper.

Let me state first that I recognize the authors' attempts to carefully consider all the critiques drawn by the Reviewers and improvement in the revised paper by following the guidance from the Reviewers. I appreciate this attitude and the collegial tone. In particular, the paper is now extended with the method from theory in the main paper.

Now I turn to the critical point on the impact of the paper. I recognize that it is difficult to create objective criteria about this issue for a given scientific journal, so thus there is a room for subjective viewpoint or taste of the Reviewers in this regard.

As this is the first report on variation of the PL signal of a single emitter upon external magnetic fields in h-BN, a 2D material, I think that the interpretation of the experimental spectra is highly critical. Direct observation of optically detected magnetic resonance would provide a solid proof that a single photon emitter with net spin has been found (that might be coherently manipulated by optical means). Here, this is not the case. Thus, in my viewpoint, the theory here is required to provide a model that can support that the observed magnetic field dependence of the signal is an inherent property of the selected single photon emitters, and one can fully exclude that the observed features are not related to some "cross-talks" or artefact in the measurements (e.g., the detectors' sensitivity is affected by the presence of the magnet, etc.). That is why I found to be important to demonstrate a convincing correspondence between the theoretical model and the measured data points in Fig. 4 (e)-(g).

Our Response: We greatly appreciate the reviewer's comments and thank them for their careful and critical reading of both our initial and revised manuscripts. We acknowledge the reviewer's desire to see theory that fully models our experimental observations, but the large parameter space of possible defect structures (both chemical and electronic) prevents this. We would like to emphasize, however, that we

have explored many possible experimental pitfalls and conducted multiple control experiments in order to eliminate the possibility of experimental artifacts.

We are absolutely confident that the defects we have identified in the main manuscript and the supporting information do indeed exhibit anisotropic optical responses to an applied magnetic field. The need to account for field-induced artifacts in the measurement is the main reason for plotting data throughout the paper as % variation in the PL. We further normalized the data for small variations in the PL collection efficiency that occur when the sample is rotated in the absence of magnetic field and also due to field-dependent shifts in the microscope alignment that equally affect the signal and background.

Action Taken: In the PL Variation with Magnetic Field Measurements section of the Methods, we have added the following: “Eliminating experimental artifacts that can indicate a false response to magnetic fields is key to determining the field-dependence of QEs in h-BN. We took steps to account for experimental artifacts including plotting the percent variation in the PL rather than the raw PL with and without a magnetic field.”

I understand that it is very challenging to observe new data points (star symbols in the figure). However, novel data coming from extra measurements could strongly support the consistency between the model and the magnetic field dependence of the PL of single photon emitters.

Our Response: We sympathize with the reviewer’s view that additional data is desirable, but would like to emphasize that measured data points in Fig. 4 (e-g) correspond to the same emitter as shown in Figs. 1-3. While there are just a few data points in Fig. 4 showing consistency with the singlet ground state model, all the data presented in Figs. 1-3 are used for comparisons to our models. Only a subset of electronic configurations give rise to field-dependent variations in steady-state emission, and the qualitative agreement between our model and experimental observations takes a significant step in narrowing the field of possible electronic structures of these field-dependent QEs. We agree that it will be crucial to obtain better statistics on field-dependent emitters moving forward, but as it stands, the observations we report make a significant and important contribution to the field of optically-addressable defects in semiconductors.

The few single photon emitters that show magnetic field dependent PL are in the class having singlet ground state and a metastable triplet state, presumably with C_{2v} symmetry. Unfortunately, almost all the defects with closed shell singlet electronic configuration have a triplet excited state beside singlet excited states. From optical characterization point of view, PLE measurements (presumably at low temperatures) may reveal special features that could help understanding why these emitters are unique in terms of magnetic field dependent PL compared to the other single photon emitters. I recognize that these measurements would again require additional efforts (that might be not simple and straightforward) from the authors. Nevertheless, that would significantly raise the impact of this study.

Our Response: We completely agree with the reviewer that low temperature PLE measurements may well indicate features which help to clarify the uniqueness of these field-dependent QEs. As the reviewer comments, such measurements would require significant additional experimental efforts and are beyond the scope of the current study.

Action Taken: In the section at the end of the manuscript where future experiments are discussed, we have included the following sentence: “Low temperature optical spectroscopy may aid in identifying optical features specific to QEs which exhibit field-dependent emission.”

All-in-all, in my opinion, this is a very interesting report on spin-dependent PL signal of single photon emitters in h-BN but, at least, the optical (and/or electrical) properties of these single photon emitters should have been more thoroughly studied, in order to raise the impact to the very high scientific standard of Nature Communications. Although, I recognize this is a rigorous viewpoint that can be considered and evaluated by the Editors of Nature Communications.

Our Response: We do appreciate the reviewer's perspective as to the scope of this paper. Future work by us and others will be necessary to fully understand and model the processes behind our observations. However we strongly believe that these results add significantly to the conversation regarding defects in low-dimensional materials by demonstrating that magnetic-field-dependent emission is indeed possible.

Reviewer #3 (Remarks to the Author):

I am happy with the author responses to my queries, with the edited manuscript and to recommend this for publication.

Our Response: We sincerely thank the reviewer for their time and effort regarding their critique of this manuscript. Their input has been extremely useful.